# Seeding African Forest and Landscape Restoration: Evaluating Native Tree Seed Systems in Four African Countries

Fiona L. Giacomini [1,*,†], John A. Prempeh [2,†], Riina Jalonen [3], Barbara Vinceti [4], Marius Ekue [5], Ennia Bosshard [6], David F. R. P. Burslem [2] and Chris J. Kettle [1,4,*]

1   Department of Environmental Systems Science, ETH Zürich, 8092 Zürich, Switzerland
2   School of Biological Sciences, University of Aberdeen, Cruickshank Building, St. Machar Drive, Aberdeen AB24 3UU, UK; j.prempeh.21@abdn.ac.uk (J.A.P.); d.burslem@abdn.ac.uk (D.F.R.P.B.)
3   Bioversity International, Serdang 43400, Malaysia; r.jalonen@cgiar.org
4   Bioversity International, 00153 Rome, Italy; b.vinceti@cgiar.org
5   Bioversity International, Yaoundé 2008, Cameroon; m.ekue@cgiar.org
6   Centre for Ecology and Conservation, University of Exeter, Stella Turk Building, Penryn TR10 9FE, UK; eb872@exeter.ac.uk
*   Correspondence: fionalisa.giacomini@outlook.com (F.L.G.); c.kettle@cgiar.org (C.J.K.)
†   These authors contributed equally to this work.

**Abstract:** Commitments to Forest and Landscape Restoration are rapidly growing and being implemented globally to tackle the climate and biodiversity crises. Restoration initiatives largely based on tree planting necessitate an increased supply of high-quality and suitably adapted tree planting material. We evaluated the native tree seed supply systems in Burkina Faso, Cameroon, Ghana, and Kenya, four countries with large commitments to increase tree cover. We applied an established indicator framework to assess the adequacy of any current tree seed system to meet national needs. The study aimed to analyse (i) how well-established the native tree seed supply systems are, (ii) how public and non-public actors differ regarding the perception of existing seed systems, and (iii) the main barriers to strengthening current seed systems. Our findings identified significant gaps in the native tree seed supply systems of the four countries, arising particularly from shortfalls in the enabling environment. We found a lack of involvement of local community members in the seed systems, with a crucial need for strengthening policy, capacity building and investment in seed systems. We propose a multi-stakeholder approach and the application of online tools to improve seed systems to meet the demand for high-quality native tree seeds.

**Keywords:** Burkina Faso; Cameroon; Ghana; Kenya; tree seed supply; native tree seed; forest and landscape restoration

## 1. Introduction

Forest and landscape restoration (FLR) has great potential to mitigate land and vegetation degradation [1] and to restore ecosystem services, including carbon sequestration. Although FLR practices can include active or passive elements [2], many national commitments focus on active FLR and, more specifically, on tree planting [3]. Ambitious global restoration targets, such as the Bonn Challenge and the UN Decade on Ecosystem Restoration, are catalysing substantial investments and political commitments to increase tree cover and plant millions or even billions of trees. However, evidence increasingly suggests that many of these interventions fail and do not provide any long-term ecological or social benefits [4].

Burkina Faso, Cameroon, Ghana, and Kenya in sub-Saharan Africa have set ambitious targets to restore a total of over 24 million hectares of degraded land by 2030 as part of the AFR100 initiative (Table 1). The four study countries are all characterised by their high biodiversity. Around 60% of the sub-Saharan African population depends on forests for

food, timber and essential services [5], which are continuing to be degraded [6]. Between 2000 and 2020, the region lost 10.4% of its forest area [7]. Given the high dependency on functioning forest ecosystems, there is an urgent need to reverse this trend and boost FLR practices.

**Table 1.** Restoration commitments of the four countries.

|  | Burkina Faso | Cameroon | Ghana | Kenya |
|---|---|---|---|---|
| Restoration commitment (AFR100) | 5 Mio ha | 12 Mio ha | 2 Mio ha | 5.1 Mio ha |

One of the difficulties of implementing successful FLR practices is the multidimensionality of the problems they aim to solve. Ecological and socioeconomic criteria need to be met whilst also meeting governmental priorities (such as food security and financial growth) [1]. The use of high-diversity plantations has been shown to improve soil carbon sequestration and increase ecological stability [8]. For successful restoration outcomes, it is important to select species that are well-adapted to local environmental conditions and the socioeconomic context and use high-quality, genetically-diverse planting material [9]. Despite the importance of planting native species, exotic tree species are predominantly planted due to availability, rapid growth, and demand for commercial or utilitarian use [10,11]. Many countries lack species and genetic diversity in tree planting [12,13]. Although exotic tree species offer livelihood benefits, they support less biodiversity than native tree species [14–16].

Scaling the delivery of high-quality and diverse native tree seeds is thus an important factor in maximising the impact of restoration initiatives. The quantity and quality of native tree seeds need to be considered to achieve the ambitious global restoration targets, especially in restoration across developing countries where local needs are of high priority. The population size of the mother trees from which seeds are collected is critical, as it influences genetic diversity and progeny performance [17,18]. Best practice recommendations on seed collection emphasise the need to collect seeds from populations of at least 50 mature individuals per species and from at least 30 widely-spaced trees in each population [19–21]. However, these recommendations are rarely considered in restoration interventions on the ground, resulting in a lack of genetic diversity in the restored forest landscapes [11,22–27]. Establishing restored populations with inbred seeds and seeds of narrow genetic diversity results in slow growth, poor survival, low reproductive success and natural regeneration in future generations, as well as reduced capacity to respond to environmental changes [28]. High-quality, genetically diverse seeds are needed for a functioning, resilient and sustaining ecosystem in the future [29,30].

In this context, the seed supply system represents the set of activities, capacities and institutions required to deliver forest reproductive material to FLR projects and includes the collection, production, distribution, and quality control of seeds, seedlings, and cuttings [31]. Many countries lack sector-level resources such as seed zone maps or registries of seed suppliers that would help projects plan seed-sourcing strategies. Where such resources exist, they are not always easily accessible, especially to actors outside the public domain. This is important, given the rapid growth of diverse FLR actors who may lack capacities to select and acquire suitable seeds. At the same time, improving access to existing resources provides low-hanging fruit for improving capacities sector-wide. For sector-wide capacity enhancement, the involvement of local communities is an important factor [9,32]. Research in South America and Asia indicates a significant lack of such involvement [33–35]. Furthermore, many countries also lack institutional arrangements and technical capacities to develop and maintain a fit-for-purpose seed supply system [11,27,31,33,36–38]. This highlights the need to identify barriers in national seed supply systems and their root causes. Additionally, there is a significant knowledge gap across the African continent.

This study provides insights into the current strengths and weaknesses of native tree seed supply systems for FLR in Burkina Faso, Cameroon, Ghana, and Kenya and identifies

main capacity needs and opportunities for improvement. Building on an established indicator system, we analyse (i) how well established the native tree seed supply systems are, (ii) whether or not public and non-public actors' perceptions of existing seed system capacities differ, and (iii) the main barriers to strengthening existing seed systems.

## 2. Materials and Methods

### 2.1. Focus Countries

Based on their above-mentioned ambitious FLR commitments, the study focuses on Burkina Faso, Cameroon, Ghana, and Kenya.

### 2.2. Analysis of the Tree Seed Systems

To gain insights into the seed supply systems and allow cross-country comparisons, an expert survey was carried out using a methodology adapted from Atkinson et al. [22], based on a set of questions covering 16 indicators that collectively identify strengths and weaknesses in the seed supply system. The indicators were grouped into five components: (i) selection and innovation; (ii) seed harvesting and production; (iii) market access, supply and demand; (iv) quality control; and (v) enabling environment. An additional indicator was added to the indicator set to assess the involvement of local community members due to the high importance of their inclusion. The additional indicator, "involvement of local communities", adopted for this study was embedded in the enabling environment (Supplementary Material Table S1).

In Burkina Faso, Cameroon and Kenya, the survey was carried out using the online tool Survey Monkey. The online questionnaire was sent to experts from August 2022 to October 2022. Experts were defined as stakeholders working in the seed supply systems, largely in government and research institutions, as non-public stakeholders were more difficult to reach and less responsive. They were identified through online searches targeting information on restoration projects, through networking with researchers in each country, and by inviting responses through social media. To engage local actors, online phone interviews were organised in Burkina Faso and Kenya, and face-to-face interviews in Cameroon. The survey consisted of a multiple-choice questionnaire on the characteristics of any existing tree seed system, where each question had answering options with the scores 0 = not existent, 1 = incipient, 2 = moderate, and 3 = entirely covered, or "I don't know". The score for each indicator was then rescaled from 0 to 10, as suggested by Atkinson et al. [22], with all questions carrying equal weight. In this way, a macro-indicator for the five components was also calculated, applying the same sampling approach and analysis according to Bosshard et al. [27]. This was performed by dividing the achieved score by the maximum possible score and was then multiplied by ten.

The approach was identical in Ghana, except that the online survey was created during the period from May to June 2022, and the questionnaire did not include the "I don't know" option, the additional indicator targeting local community involvement and the open-ended question about the main challenges faced by actors in the context of restoration activities. The sample size for Ghana was larger (100) than the sample size in the other three countries (20, 20, and 28) because people were approached on-site and not just online regarding filling out the questionnaire.

The indicator framework used generates insights into the stakeholders' perceptions about existing capacities within the tree seed systems. Capacities in this article are defined to include technical capacities, institutional arrangements, and the wider enabling environment [39]. The enabling environment entails the broad social system within which people and organisations function [39]. It encompasses legal support, political will, adequate investment in research activities and capacity-building programmes to enhance the seed supply system. The enabling environment sets the overall scope for capacity development [39].

### 2.3. Sub-Divisions of the Analysis

In order to address the research question asking if different perceptions of current capacities exist between public- and non-public stakeholders, the answers were separated into the public (62%) and the non-public (38%) sectors. The public was defined as an organisation/institution within the state/government structure, organised by the government and financed by public funds. Non-public was defined as being primarily privately financed and includes actors such as civil society, community organisations and businesses. The differentiation between public and non-public can be seen in Table 2. In Ghana, 55 of the 100 responses could be classified as public and non-public, while the remaining 45 respondents did not disclose this information and their responses have been excluded from the sample used to address this question.

**Table 2.** Organisation types corresponding to the public and non-public sectors.

| Public | Non-Public |
|---|---|
| • Academia (publicly financed university)<br>• Intergovernmental organisation<br>• National institution or agency | • Civil society or NGO<br>• Farmer or producer association<br>• Local community<br>• Private company |

### 3. Results

A total of 168 completed questionnaires were collected, consisting of 20 from Burkina Faso, 20 from Cameroon, 100 from Ghana, and 28 from Kenya. Data were collected from experts working in public institutions (37%), private agencies (13%), civil societies or NGOs (8%), academia (7%), local communities (6%), farmer or producer organisations (1%), intergovernmental organisations (1%), and unknown (27%).

The comparative analysis of the macro-indicators for the four countries, namely (i) selection and innovation, (ii) seed harvesting and production, (iii) market access, supply and demand, (iv) quality control, and (v) enabling environment are similar among the four countries and show that basic capacities for the seed systems exist, even if they are rather insufficient (Figure 1).

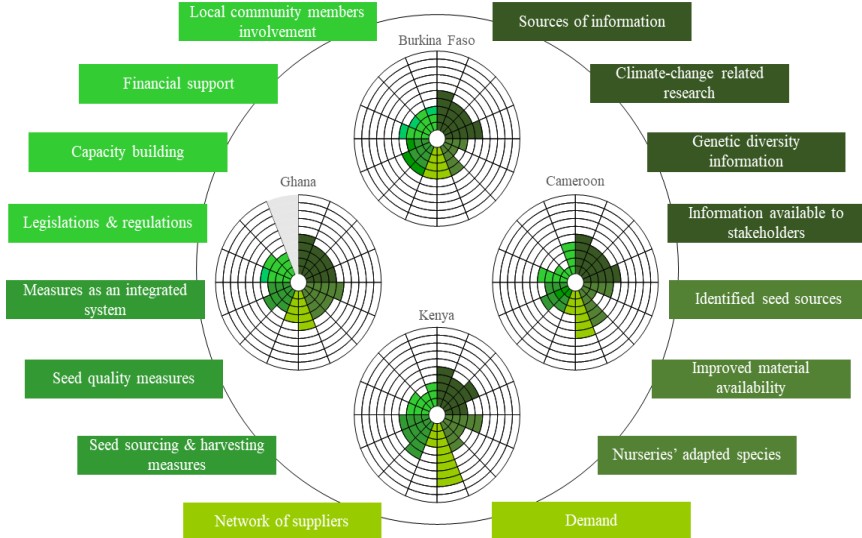

**Figure 1.** Indicator scores by country. The text boxes represent the titles for each indicator, shown clockwise as the bars of the data in the radial diagram for each country. Shaded segments indicate the weighted values of scores from questionnaire responses on a scale from 1 (low support) to 10 (maximum support) for each indicator. The grey segment on the chart for Ghana indicates where data were not collected.

### 3.1. Comparing Tree Seed Systems across the Four Target Countries

The assessment showed that while national seed systems exist in the four countries, they are insufficient across the five macro-indicators. The following section provides a comparative overview of the key findings among the four countries.

Seed selection and innovation: Seed selection and related innovations were evaluated as more effective by Burkinabe and Cameroonian respondents than by Ghanaian and Kenyan respondents. For seed accessibility, all respondents except Ghanaians perceived information to be less accessible, or they did not know if information existed at all. Seed harvesting and production: Seed-sourcing systems were evaluated as more effective across the three indicators in Ghana, although still rated as inadequate. In Burkina Faso and Cameroon, respondents reported poor identification of seed sources, and in both countries, as well as Kenya, improved material was considered lacking. Market access, supply and demand: In all countries, the results indicated a clear demand for priority native species and suitable provenances, except for Burkina Faso, where demand for suitable provenance was low. In all countries, but especially Kenya and Cameroon, respondents highlighted a lack of functioning seed suppliers' networks. Quality control: High-quality and certified seeds of native tree species are generally lacking in all countries, although they are somewhat better developed in Kenya, where a certification system is planned within the new Seed Act, among others. However, this Seed Act still has to be enforced. Enabling environment: There are no policies explicitly targeting native tree species for restoration in any survey country. Respondents also emphasised that capacity building and financial support are largely unavailable, and the involvement of local community members in seed supply chains is very limited in the three countries where this was examined. The scoring of the five macro-indicators across the four countries can be found in the Supplementary Material Table S2.

### 3.2. Differences between Public and Non-Public Respondents

In addition to the differences between countries, differences were also found between public and non-public stakeholders' perceptions and knowledge of the seed systems within their respective countries. In Cameroon, Burkina Faso and Kenya, non-public respondents had less knowledge of (higher "I don't know" rate) and less awareness of existing seed system resources: around twice as many "I don't know" answers were provided by non-public experts ($39 \pm 7\%$) compared to the public experts ($23 \pm 16\%$). The "I don't know" rate from the non-public experts was highest in the Quality Control macro-indicator ($64 \pm 15\%$). Even after excluding all "I don't know" answers from the analysis, a discrepancy in knowledge was found between public and non-public actors.

All seed supply systems' capacity indicators were assessed by non-public experts as equally low or weaker than by public experts (Table 3). One exception is the macro-indicator of seed harvesting and production in Cameroon, where non-public respondents generally rated this as higher. This difference for one indicator in Cameroon mainly results from the indicator for the availability of improved material, where non-public actors perceived the availability as being higher than the public actors. Results of the macro-indicator for market access, supply and demand were similar between public and non-public actors in Burkina Faso, Cameroon and Kenya, the latter having higher scores. The higher scores can be explained by the very high demand for native tree seeds. Quality control indicators were rated relatively highly in Kenya and Burkina Faso by public stakeholders, but here again, non-public stakeholders' perceptions of these indicators were much lower. All other results of the remaining sub-groups showed that quality control is weak. The enabling environment indicators were generally perceived as weak or very weak among all sub-groups and were perceived as lower by non-public stakeholders in all countries except Kenya.

**Table 3.** Different perceptions of the existing capacities in the five macro-indicators. Average results of public and non-public actors with standard deviation. Scores can range from 0 (non-existent) to 10 (fully existent).

| | Burkina Faso | | Cameroon | | Ghana | | Kenya | |
|---|---|---|---|---|---|---|---|---|
| | Public (n = 4) | Non-Public (n = 16) | Public (n = 15) | Non-Public (n = 5) | Public (n = 48) | Non-Public (n = 7) | Public (n = 9) | Non-Public (n = 19) |
| Selection and innovation | 4.8 ± 0.4 | 4.5 ± 0.8 | 5.2 ± 0.5 | 3.6 ± 0.7 | 4.6 ± 0.4 | 4.0 ± 0.4 | 4.8 ± 0.9 | 4.0 ± 0.8 |
| Seed harvesting and production | 5.2 ± 1.5 | 3.0 ± 0.8 | 3.2 ± 1.4 | 4.7 ± 0.6 | 4.9 ± 0.2 | 4.6 ± 0.1 | 5.6 ± 0.2 | 3.3 ± 1.0 |
| Market access, supply and demand | 4.4 ± 0.0 | 4.3 ± 0.2 | 4.2 ± 1.3 | 4.2 ± 2.5 | 5.5 ± 0.3 | 3.7 ± 0.6 | 5.3 ± 3.6 | 5.1 ± 1.7 |
| Quality control | 6.1 ± 0.5 | 2.6 ± 0.5 | 4.1 ± 0.7 | 0.0 ± 0.0 | 3.4 ± 0.3 | 3.2 ± 0.1 | 6.5 ± 0.6 | 3.3 ± 0.5 |
| Enabling environment | 3.8 ± 0.5 | 2.9 ± 0.1 | 3.0 ± 1.7 | 2.0 ± 0.4 | 4.0 ± 0.6 | 3.3 ± 0.3 | 3.1 ± 0.5 | 2.7 ± 0.6 |

*3.3. Comparing the Five Macro-Indicators across the Four Target Countries*

3.3.1. Selection and Innovation in Seed Sources

Survey results showed that lists of priority native species, including threatened species, have been compiled in all four target countries. For example, the National Tree Seed Centre (CNSF) in Burkina Faso developed a catalogue of native and exotic tree species for which they produce planting material, with seed sourced across six provenance regions. Research on the effects of climate change on native species appears to be limited in Burkina Faso and more evident to respondents in Cameroon and Ghana, but many Kenyan stakeholders did not know if climate-change-related research existed or not. The results suggested that provenance trials are more frequent in Ghana than in the other countries, and Burkina Faso and Kenya appear to have provenance trials for only a few priority native species. Results from Cameroon show that many respondents are unaware of provenance trials. The accessibility of information about selection and innovation for stakeholders is limited in Burkina Faso, Ghana and Kenya. In Cameroon, there was no agreement on the accessibility of information among the respondents. In all four countries, non-public stakeholders perceived the accessibility of resources for seed selection to be lower than public stakeholders did.

3.3.2. Tree Seed Harvesting and Production

Despite the identification of some seed sources, the protection of these seed sources appeared to be insufficient across all four countries. Results from Ghana showed that tree seed sources receive some protection, including over 266 gazetted forest reserves. Kenyan respondents showed no agreement on the protection of tree seed sources. Improved material appeared to be mainly lacking in Burkina Faso and Cameroon and was perceived to be most available by Ghanaian respondents. In Burkina Faso, respondents from the non-public sector perceived that seed production was more limiting than respondents from the public sector, while in Cameroon, public-sector actors perceived production to be lower.

3.3.3. Market Access, Supply and Demand for Tree Seeds

Results showed that there is a perceived demand for priority native species in all four countries, with a particularly high demand in Kenya, followed by Cameroon and then Ghana. Demand for suitable provenances seemed to be very limited in Burkina Faso, and often, information on specific characteristics of different provenances is not provided. Ghana and Kenya appeared to have a higher demand for information on seed characteristics, which is mainly expressed by public-sector actors. Responses from the four target countries were dissimilar in relation to the existence of a planting-material suppliers' network at the national, regional, or local level. The lack of such a network was particularly evident among responses from Cameroon and Kenya. A functioning, sustaining seed network is perceived to be missing by respondents in these countries.

3.3.4. Quality Control of Native Tree Seed

In Burkina Faso, there was a large discrepancy of views among respondents from the public and non-public sectors regarding this macro-indicator, with the latter group perceiving a greater capacity in quality control. Half of the respondents (all from the

non-public sector) appeared not to know if a certification system for native tree seeds exists or not. The other half, mainly from the public sector, indicated that a certification system exists. None of the Cameroonian non-public stakeholders showed knowledge of a certification system. Most surveyed Ghanaian actors stated that there is a certification system for native tree seeds, but it does not cover all aspects of seed sourcing, harvesting and production. Non-public respondents perceived the certification system to cover fewer aspects of the seed supply system than respondents from the public sector. This was also observed in Kenya, as public actors tended to argue that the measures were better covered. However, there was no concordance of the answers in Kenya.

### 3.3.5. Enabling Environment for Effective Native Tree Seed Systems

In the indicator about the enabling environment, half of the Burkinabe respondents (most from the non-public sector) were not aware of any policies for native tree seed supply. Results from Cameroon, Ghana and Kenya showed that some policies exist, but their execution and implementation appear to be weak. Capacity building and financial support seemed to be lacking across all analysed countries but were perceived to be stronger by respondents from Ghana than the other three countries. Local community members were perceived to be included to some extent when it comes to the identification, protection and management of tree seed sources in Cameroon and Kenya (not assessed in Ghana). In contrast, involvement was shown to be mainly lacking in the elaboration of policies and in the access to seed certification. Burkinabe respondents from the public sector perceived the involvement of local community members to be higher than respondents from the non-public sector. Overall, local community members did not seem to be sufficiently involved in the tree seed supply chain in any of the three countries where this indicator was assessed.

## 4. Discussion

### 4.1. Are Current Native Tree Seed Systems Sufficient to Meet the Demand?

This article provides insights into existing capacities underpinning the native tree seed supply system for FLR in Burkina Faso, Cameroon, Ghana and Kenya. Using an online survey and interviews, the existing native tree seed supply systems were analysed. We also examined how public and non-public stakeholders differ in their perceptions, partly because it is increasingly recognised that to scale production of native tree planting material will require a cross-sectoral approach. Our results showed that while basic systems exist in all four target countries, they are insufficient to meet demand (Figure 1). Across the four countries, the enabling environment had the lowest scoring. The shortfall in national capacity in all countries indicates that the high demand for native tree seeds to achieve the countries' ambitious restoration targets may not be met. This demand is expected to rise even more in the near future because of restoration targets that are aimed to be met within the first half of this century [22], which emphasises the importance of investment in the national seed systems.

A shortage of high-quality seed and planting material was noted in particular by respondents from Burkina Faso and Cameroon. This could be due to the low questionnaire response rate from these two countries. Even where demand is high, it cannot be met either because of the low volume of quality seeds, a lack of suitable provenances, or both. This analysis also showed that quality assurance mechanisms are widely lacking in Burkina Faso, Cameroon, and Ghana. Similarly, studies from other African and Asian countries suggest that the market for native seeds is generally weak, and certification systems are lacking [27,31]. The results emphasised that seed supply systems with suitable provenance need to be greatly strengthened. For example, in its National Forest Programme 2016–2030, Kenya aims to sustainably restore forests through a cross-sectoral and multi-stakeholder framework [40]. In Ghana, the Ghana Forest Plantation Strategy [41] sets a focus on producing high-quality seeds and is the recent umbrella project.

Understanding the potential effects of climate change on native tree species is necessary for the long-lasting success of FLR. Through climate change, local sources may not be adapted to the changing growth conditions, especially as areas targeted for restoration are often degraded, and existing populations may possess a low number of individuals and lack genetic diversity. Thus, alternative provenances that match future predicted climates for each restoration site need to be considered [42]. This needs to be studied, as many interviews and survey responses indicate that seed is very often collected locally. Tools such as MyFarmTree (www.myfarmtrees.org (accessed on 10 July 2023)) and Diversity for Restoration (www.diversityforrestoration.org (accessed on 10 July 2023)) can digitise data collection and help with the decision-making of suitability of species and provenances. Our results showed that information related to climate change and its impacts on seed-sourcing strategies are very limited in all four countries, which is similar to the findings of a similar study of four Asian countries [27].

### 4.2. Importance of Empowering Local Communities to Contribute to Seed Systems

Currently, the enabling environment is insufficient for a fit-for-purpose tree seed system in the four analysed countries. This is the sole macro-indicator where all public and non-public actors assessed these aspects of the seed system as weak. The enabling environment needs to include (i) policies, (ii) active capacity building, (iii) financial investment in seed systems [33], and (iv) empowerment of local community members. These conditions represent a necessary platform for strengthening the other components of seed supply systems. Technical and entrepreneurial training on seed production and marketing is needed for local community members. Due to their proximity to the remaining seed sources and restoration sites, they can play key roles in native tree seed production while benefiting from employment and income opportunities [35,43]. In previous African and Brazilian studies, inadequate financial investments for restoration capacities were identified as a constraint to successfully producing local, high-quality native tree seeds [43,44]. Implementers and funders of restoration need to ensure that adequate funds are allocated to sourcing quality seeds as the biological foundation for restoring viable, productive and resilient ecosystems.

Our results have shown that whilst public and non-public actors' perceptions of seed supply systems were often similar, non-public actors were more likely to perceive existing systems as relatively weaker compared to the public stakeholders (Table 3). Non-public actors are, among others, representatives of local communities and the informal system, as they can be entirely privately financed and may not be officially registered. Research in Brazil, Ethiopia and Uganda has shown that local community members are the main actors in informal seed systems, which often dominate seed supply [26,37,45], at least for crop seed systems. Generally, the formal system (actors operating under the regulatory control of the state) is distinguished from the informal system (actors operating outside of the state's regulatory control, e.g., informal associations or individual stakeholders). In practice, those two models are often inter-connected. The formal sector often collaborates with the informal sector to obtain seeds and seedlings, while informal actors may depend on seed sources managed by the formal sector. Seed supply systems vary from country to country, depending on the country's history, the government, the policies, the infrastructure and the enabling environment. Informal seed systems play an important role in the provision of tree seed for FLR, particularly where the formal system is weak but is poorly documented. Besides the different perceptions of the strength of existing capacities, the "I don't know" option was selected more often by non-public respondents. A lack of cooperation among all stakeholders engaged in seed systems weakens the supply of native tree seeds [4,46,47]. Therefore, a multi-stakeholder approach where the different sectors work together is important for FLR. As local community members are often the actors who implement FLR projects [9,35], providing incentives and training for these local groups is critical. Urzedo et al. [43] showed that community networks promote multiple stakeholders' engagement. It is important to understand in detail the local community members and

ensure that all groups, including men, women and youth, are well represented to achieve people-centred FLR [48].

The second important aspect to consider is the lack of participation possibilities in the past, namely that there is no culture of participation, which needs to be considered when collaborating with different sectors. Accessibility to training and rural resources needs to be tailored to local needs, as this has been found to be a limiting factor when trying to involve local community members in FLR [35,37].

Notable projects implementing community-led seed systems include the multi-stakeholder coalition Atlantic Forest Restoration Pact (AFRP) and the Xingu Seeds Network in Brazil, where actors from different sectors have worked together to develop solutions that address the diverse needs of the target area [43,49]. The lessons learned from the AFRP, the Xingu Seeds Network, and other similar projects can be used and implemented for other initiatives of this kind. A good example of such learning is the need to support business-skills training for local community associations and a flexible bottom-up governance approach.

*4.3. Opportunities to Scaling Effective Native Tree Seed Systems*

While this assessment of the seed systems in Burkina Faso, Cameroon, Ghana and Kenya found barriers across all five macro-indicators, the most urgent barrier is the inadequate enabling environment. We recommend establishing an appropriate restoration-practice framework with appropriate funding and well-established training for seed collection, seed tree conservation and nursery practices, together with economic incentives for engaging in rural entrepreneurial nurseries. This is also supported by other authors as a first step towards a stable foundation [26,50]. Establishing a seed supplier network would then additionally help to meet the demand for native tree seeds [51]. A collaboration between public and non-public sectors is important so that the seed supply chain can meet the demand for high-quality native tree seeds [52]. Connecting community-based networks with public forest offices poses a great opportunity. Substantial financial investment is needed to create a successful seed supplier network.

Given this urgent need to scale the delivery of native tree seeds in all countries, it is important to develop solutions that consider multiple dimensions. It would be useful to evaluate both the current state of seed supply systems and also practices used in the past (the country's history). This will inform forecasting the climate-change impacts on forest genetic resources and their availability [53]. Certain elements will be critical for scaling the delivery of native-tree seeds of native-tree seeds in Africa, as highlighted by Bosshard et al. [27] for Asia. This will include developing digital infrastructure and building capacity to use existing resources accessible to rural communities. Such resources may include seed zones, species selection (www.diversityforrestoration.org (accessed on 10 July 2023)), or digital tools for documentation and verification of seed systems (www.myfarmtrees.org (accessed on 10 July 2023)), as already mentioned above.

**5. Conclusions**

To achieve their combined commitments to restore 24.1 million hectares of degraded land, Burkina Faso, Cameroon, Ghana and Kenya need functioning seed supply systems. These will respond to the demands for high-quality tree planting material for native species. Our assessment has revealed that basic seed system infrastructures exist in all four countries, but these are insufficient to meet their ambitious restoration targets. The experts surveyed for this study indicated that whilst barriers to an adequate seed supply vary across countries, the lack of an enabling environment (namely, a lack of policies, capacity building, financial investment and involvement of local community members) is the main challenge in all countries. The published literature points to a lack of representation and, thus, also of consideration of local community members in the public sector. A greater involvement of stakeholders from different societal levels and sectors would be an important step in building a better foundation for further capacity development. A suppliers' network could also help to meet current and future demand of high-quality native tree seeds through

cooperation and a distribution of efforts across multiple players. Digital tools offer a powerful solution for addressing these issues and require investment. Mobile-phone-based platforms, which are becoming increasingly widespread across Africa, provide emerging opportunities for community engagement in both formal and informal native tree seed systems. These could not only increase the volume and diversity of quality seed supply but offer additional income options for rural communities and an additional incentive to conserve native trees.

**Supplementary Materials:** The following supporting information can be downloaded at https://www.mdpi.com/article/10.3390/d15090981/s1, Table S1: Online Questionnaire; Table S2: Average results of the five macro-indicators with standard deviation.

**Author Contributions:** Conceptualization, F.L.G. and J.A.P.; methodology, F.L.G., J.A.P. and E.B.; software, F.L.G., J.A.P. and R.J.; validation, F.L.G. and J.A.P.; formal analysis, F.L.G.; investigation, F.L.G. and J.A.P.; resources, B.V., J.A.P. and M.E.; data curation, B.V., F.L.G., J.A.P. and M.E.; writing—original draft preparation, F.L.G.; writing—review and editing, B.V., C.J.K., D.F.R.P.B., E.B., J.A.P. and R.J.; visualization, F.L.G.; supervision, C.J.K. and D.F.R.P.B.; project administration, C.J.K. and D.F.R.P.B.; funding acquisition, n/a. All authors have read and agreed to the published version of the manuscript.

**Funding:** Funded by the UK Government through Darwin Initiative Grant 28-007, the Global Environmental Facility (GEF) Restoration Challenge Grant Platform for Smallholders and Communities with Blockchain-Enabled Crowdfunding and the CGIAR Nature-Positive Solutions Initiative.

**Institutional Review Board Statement:** Not applicable.

**Data Availability Statement:** Data are available upon request.

**Acknowledgments:** The authors would like to acknowledge the contribution of the respondents of both the experts' survey and the interviews and of all the stakeholders who helped to collect the data. Further, we want to thank Moussa Ouedraogo (the National Tree Seed Center of Burkina Faso) and Francis Oduor (Alliance of Bioversity and CIAT) for their help in gathering relevant information and establishing contacts with key stakeholders, Joseph Mireku Asomaning (National Tree Seed Centre, Forestry Research Institute of Ghana Kumasi), Lawrence Damnyag (FORIG—Kumasi), Kofi Affum-Baffoe & Jones Agyei Kumi (RMSC, Kumasi), John Apah (Plantation Department, FC—Headquarters, Accra) Joseph Osiakwan (Forestry, Ministry of Lands and Natural Resources, Accra) and all past students of Faculty of Forest Resouces Technology (2005–2009) batch for their immense support. Additionally, for his editorial support, the authors acknowledge Vincent Johnson, consultant editor for the Bioversity-CIAT Alliance science writing service.

**Conflicts of Interest:** The authors declare no conflict of interest.

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
