# Peer review of "Seeding African Forest and Landscape Restoration: Evaluating Native Tree Seed Systems in Four African Countries"

_diversity, doi:10.3390/d15090981_

Round 1
Reviewer 1 Report
The authors present an interesting study on the state of the seed supply chain for the propagation of native species in four African countries, in order to meet their international commitments. I think the manuscript has valuable information and can be published in its present form.
I have only two minor comments.
1. When reading the introduction and the methods, it is not clear what the authors mean by enabling factors. In lines 142-143 capacities are defined to include technical capacities, institutional arrangements, and the wider enabling environment… but still one wonders what this wider environment is. It is only in the discussion (lines 338-340) and conclusions (lines 420-421) that the reader understands what conditions define an enabling environment. It would be easier to follow their line of thoughts if they specify from the beginning what they mean by enabling environment.
2. Their sample size is rather small for Burkina Fasso and Cameroon and thus the conclusions that may be extracted are less sound than those from Ghana or Kenya, and the authors should acknowledge it in the discussion. Perhaps framing those numbers in the context of the state and/or size of the government institutions in charge of reforestation programs could help to visualize whether it is a very small sample size or not.
Author Response
We highly appreciate the comment. Changes have been made accordingly:
- Lines 144 ff.: Explanation of enabling environment
- Lines 316-317: Highlight low response rate in discussion
Please see attachment.

Reviewer 2 Report
Dear authors,
I very much enjoyed having the opportunity to read your work. Congratulations on the research developed on an important topic.
To make the study more comprehensive, the authors could provide information on each country's most important forest species and the main restoration objectives. Presenting an additional table with this information would make the work more complete.
Author Response
We appreciate the reviews comment but do not think it will be helpful to provide a full list of forestry tree species in the paper as we already mention that species lists are available through online tools such as Diversity for Restoration. In addition given the broad and diverse objectives for restoration we are not able to provide generalisations at the country level, as this was not part of the questionnaire. We highlight here that we do include in the discussion ‘This will include developing digital infrastructure and building capacity to use existing resources accessible to rural communities. Such resources may include seed zones, species selection (www.diversityforrestoration.org), or digital tools for documentation and verification of seed systems (www.myfarmtrees.org), as already mentioned above.’